# Bioaccessibility and Bioavailability of Minerals in Relation to a Healthy Gut Microbiome

**DOI:** 10.3390/ijms22136803

**Published:** 2021-06-24

**Authors:** Viktor Bielik, Martin Kolisek

**Affiliations:** 1Department of Biological and Medical Science, Faculty of Physical Education and Sport, Comenius University in Bratislava, 81469 Bratislava, Slovakia; 2Biomedical Center Martin, Jessenius Faculty of Medicine in Martin, Comenius University in Bratislava, 03601 Martin, Slovakia; martin.kolisek@uniba.sk

**Keywords:** micronutrient, trace element, physical fitness, gut microbiota, magnesium, Fe deficiency

## Abstract

Adequate amounts of a wide range of micronutrients are needed by body tissues to maintain health. Dietary intake must be sufficient to meet these micronutrient requirements. Mineral deficiency does not seem to be the result of a physically active life or of athletic training but is more likely to arise from disturbances in the quality and quantity of ingested food. The lack of some minerals in the body appears to be symbolic of the modern era reflecting either the excessive intake of empty calories or a negative energy balance from drastic weight-loss diets. Several animal studies provide convincing evidence for an association between dietary micronutrient availability and microbial composition in the gut. However, the influence of human gut microbiota on the bioaccessibility and bioavailability of trace elements in human food has rarely been studied. Bacteria play a role by effecting mineral bioavailability and bioaccessibility, which are further increased through the fermentation of cereals and the soaking and germination of crops. Moreover, probiotics have a positive effect on iron, calcium, selenium, and zinc in relation to gut microbiome composition and metabolism. The current literature reveals the beneficial effects of bacteria on mineral bioaccessibility and bioavailability in supporting both the human gut microbiome and overall health. This review focuses on interactions between the gut microbiota and several minerals in sport nutrition, as related to a physically active lifestyle.

## 1. Introduction

Physical activity, athletic training, or sport performance, depending on intensity, can increase the rate of energy turnover in skeletal muscle by up to 20–100 times that of the resting rate [1]. Therefore, higher energy and macronutrient intake is needed to meet the losses from energy expenditure, replenish glycogen stores, build muscles and repair tissues [2]. Suitable key recommendations and dietary strategies have been made with regard to energy and macronutrients for endurance athletes [3,4], game sports [5,6], strength and power sports [7,8], and track and field athletes [9].

For normal health to be maintained, a wide range of micronutrients such as vitamins, minerals, and trace elements must be present in sufficient amounts in body tissues, and dietary intake must be adequate to meet such needs [1]. However, the daily recommended need for micronutrients may not be related to the energy expenditure of physical activity [10]. Scientific data did not confirm higher micronutrient requirements in young athletes when compared to healthy controls [11]. Some researchers have determined that athletes require a higher intake of vitamins and minerals than their sedentary counterparts, whereas other researchers have reported no greater micronutrient requirements [12]. Exercise and athletic training involve the evaporation of perspiration from the skin with a sweat rate of more than 2 L/h [13]. The concentration of sodium (Na), the main component of sweat, and the daily loss of Na because of athletic training vary according to sport intensity [14], gender [15], and intra/interindividual variability [16]. However, even with prolonged periods of heavy evaporation, overall sweat-induced deficiencies seem to be of minimal importance for trace minerals and vitamins [17]. Nevertheless, some micronutrient deficiencies might occur as a consequence of an inadequate and/or imbalanced diet with poor-quality food. Interestingly, excessive energy intake does not yet guarantee a sufficient daily recommended intake of micronutrients [18].

From the perspective of the gut microbiome, certain factors are of paramount importance, such as their bioavailability, food digestion, and absorption. Our aim here is to address this point and to provide up to date evidence for nutritionists, dieticians, sport medicine physicians, and individuals with a healthy lifestyle. In this narrative review, we discuss the interaction between gut microbiome and minerals (Table 1) and explain mechanism of their bioaccessibility and bioavailability related to the gut microbiome (Table 2). We focus on major essential minerals that are commonly accessible on the market as nutritional supplements, namely calcium (Ca), magnesium (Mg), iron (Fe), zinc (Zn), and selenium (Se) (Figure 1). Particular attention will be paid to fermentation, which is one of the oldest and most economical methods used in food preservation, as a possible means of attaining micronutrient requirements for athletic and health-related purposes. The extant literature has been searched using Scholar, PubMed, Web-of-Science, and Scopus from inception to March 2021 via database-adapted search strings based on the following key-words: micronutrient, gut microbiome, physical activity, athletes, fitness, and combinations thereof.

## 2. Gut Microbiota, Mineral Availability, and Function

### 2.1. Calcium

Calcium is the most abundant mineral in the human body, making up 1.5 to 2% of total body weight. Approximately 1200 g Ca is present in the body of an adult human, more than 99% of this amount being found in bones [19,20]. Ca is a key mineral for athletes, mainly because of its roles in maintaining bone health. Insufficient delivery of Ca to the body can cause bone loss and lead to risks of bone damage and injury [30] Muscle-skeletal structures respond positively to the weight-bearing and impact-loading aspects imposed by sport activity during childhood and adolescence [31]. The muscle strength of young athletes is associated with bone mineral density and content [32]. Further, strong evidence suggests that exercise benefits bone health at all ages and is a critical factor in osteoporosis prevention and treatment in both sexes [33]. However, female athletes under exhaustive training are particularly at risk of Ca deficiencies and of development of the female athlete triad, which refers to the interrelatedness of energy availability, menstrual function, and bone mineral density [34]. An omnivorous diet can provide adequate amounts of Ca, especially when dairy products [milk, cheese, and yogurt], spinach, kale, okra, collards, soybeans, and white beans are consumed [35]. However, athletes commonly experience low energy availabilities that often result in reduced bone formation and strength and increased risk for stress fracture injuries [36]. To find the effect of instant low energy intake on bone metabolism in physically active human, Papageorgiou et al. [37] performed a randomized study with 5-day protocols of a controlled (45 kcal·kgLBM^−1^·d^−1^) and a restricted [15 kcal·kgLBM^−1^·d^−1^] diet. They concluded that five days of low energy availability decreased bone formation in both sexes and increased, but only in women, bone resorption.

Therefore, the absorption of Ca^2+^ in the intestine is essential for maintaining bone health and Ca homeostasis. Absorption occurs in transcellular and paracellular pathways [38]. The pivotal role for Ca^2+^ absorption (influx to enterocytes) in gastrointestinal tract play apically localized Ca^2+^-conducting channels TRPV6 (transient receptor potential vaniloid member 6), and perhaps to less extent also Ca^2+^-conducting channels TRPV5; and baso-laterally localized Na^+^/Ca^2+^ exchanger NCX1 and Ca^2+^ ATPase (PMCA1b), both responsible for Ca^2+^ efflux from enterocytes [39,40]. Knowledge of the molecular details of intestinal Ca^2+^ absorption might thus be essential for the prevention of osteoporosis and other pathologies related to Ca^2+^ metabolism and also for the development of nutritional and medical strategies [38]. The close relationship between cells of the immune system and bones explains the essential role of the intestinal microbiome in maintaining bone health and mineralization. Gut microbiota can increase Ca^2+^ absorption and modulate the production of gut serotonin, which is believed to interact with bone cells and to regulate bone metabolism [41].

#### 2.1.1. Effect of Prebiotics on Calcium Absorption

The influence of gut microbiota on bone and the ways in which such effects are modulated by the diet are beginning to be explored. Several types of prebiotic plant fibers that reach the lower intestine result in an altered gut microbiome through the production of short-chain fatty acids. These changes positively associated with increases in fractional calcium absorption in adolescents and with increases in measurements of bone density and strength in animal models [42]. Whisner et al. [19] investigated the effect of soluble maize fiber on Ca^2+^ absorption and retention in pubertal children. A moderate daily intake of soluble maize fiber, a well-tolerated prebiotic, resulted in increased Ca^2+^ absorption during a 3-week period of metabolic balance testing in adolescents consuming less than the recommended amounts of Ca (diet contained 600 mg Ca) [19]. When children consumed a soluble maize fiber, lower abundance of *Firmicutes* and higher abundance of *Bacteroidetes* were detected in the gut microbiota. Moreover, Ca^2+^ absorption was positively correlated with the genera *Bacteroides*, *Butyricicoccus*, *Oscillibacter*, and *Dialister* [19]. The same authors, in another study, proposed that the bacterial fermentation of soluble corn fiber to short-chain fatty acids (SCFA) reduced the gut luminal pH, which increased Ca^2+^ solubility and transcellular absorption [20]. The proportion of the genus *Parabacteroides* significantly increased with soluble corn fiber dose. The authors suggested that two groups of bacteria were involved, namely the *Bacteroidetes* (*Parabacteroides*) fermenting soluble corn fiber to acetate or lactate, and the *Firmicutes* (*Clostridium*) further fermenting these substrates to butyrate, thereby promoting increased calcium absorption [20]. The production of SCFA increases acidity in the colon. This is thought to occur by direct SCFA acidification and by butyrate activation of H^+^/Ca^2+^ exchange. The lowered pH helps to increase mineral solubility making Ca^2+^ more absorbable [43,44]. On the other hand, Ca itself can provide a prebiotic effect. Ca-fed mice exhibit increased levels of *Bifidobacterium spp.* and *Bacteroides* to *Prevotella* [45]. Currently, there is very limited amount of inconclusive molecular data addressing possible role of microbiome in the regulation of transporters involved in transcellular Ca^2+^ absorption in intestine.

#### 2.1.2. Effect of Probiotics on Calcium Absorption

Fermented dairy products (yogurts or soft cheese) have been used for thousands of years to preserve milk, extend shelf-life, and increase digestibility via lactose breakdown during fermentation. This processing of milk can be dated back to the sixth millennium BC when early agriculture began to appear in Northern Europe [46]. In humans, fermented dairy products are the primary source of probiotics [47]. Convincing evidence has been presented supporting the idea that probiotics, in dependence on the composition and metabolism of the gut microbiome, improve bone health by modulating both bone resorption by osteoclasts and bone formation by osteoblasts [48]. A few observational studies have confirmed positive associations between bone traits and the consumption of fermented milk products [49,50,51]. Even a short 2-week intervention [400 g/d] of the intake of fermented dairy products increases the abundance of *Bifidobacterium* species, *Lactobacillus delbrueckii subsp. bulgaricus*, and *Streptococcus salivarius subsp. thermophilus* [52]. Interestingly, increases in the density of lactic acid bacteria were no different following the consumption of fresh or pasteurized yogurt [53]. One possible mechanism for increasing Ca^2+^ availability is the higher absorption of Ca^2+^ in the intestine by probiotics and fermentation [54]. Gilman and Cashman [55] have previously reported that, in human intestinal-like Caco-2 cells in culture, *Lactobacillus salivarius* can increase Ca^2+^ uptake, although Ca^2+^ transport is unaffected by exposure of Caco-2 cells to probiotics [55].

The interplay between the immune and bone systems is known to be close, and chronic inflammatory conditions are associated with bone health [56]. The beneficial effect of probiotics on inflammatory conditions is well documented. The probiotic strain *Lactobacillus casei* inhibits pro-inflammatory cytokines TNFα and IL-6 and enhances anti-inflammatory cytokine IL-10 [21]. The quantity of gut pro-inflammatory bacteria is associated with the plasma levels of cytokines such as IL-6 and IL-8 and, therefore, with systemic low-grade inflammation [57]. The way that probiotics impact human bone health is an area of intense research and is of interest to the general public. Probiotics in the form of yoghurt and other fermented food items might serve as easily accessible alternatives for a healthy diet and strong bones.

### 2.2. Magnesium

Magnesium (Mg) is the second most predominant cation within cells and is crucial both for the functions of a plethora of enzymes and for neuromuscular transmission [58]. It is a mandatory mineral that is involved in hundreds of biochemical reactions and physiological functions in the body. In addition to maintaining normal nerve and muscle function, heart rhythm, vasomotor tone, blood pressure, the immune system, bone integrity, and blood glucose levels, Mg^2+^ (ionized Mg) antagonizes Ca^2+^ absorption, and thus (Mg^2+^) influences Ca homeostasis [59]. When adjusted for energy, vitamin D, Ca, and phosphorus intake, Mg is a significant and strong predictor of bone mineral density [60]. However, numerous studies report a lower Mg intake among athletes in a variety of sports [61,62,63,64,65]. Exhaustive exercise apparently increases Mg losses via urine and sweat, possibly increasing its requirements by up to 20% [66]. A lower consumption of Mg can attenuate immunity to chronic inflammatory responses with consequences for the short- and long-term health and performance of athletes [59]. In Mg-deficient individuals, an increased dietary intake of Mg or supplementation might have beneficial effects on exercise performance. However, additional Mg supplementation has not been shown to improve physical performance in physically active individuals with an adequate Mg intake [66]. Similarly, as concluded in the meta-analysis and systematic review of Wang et al. [67], the current evidence does not support an advantageous effect of Mg supplementation on muscle fitness or on physically active individuals with an adequate or relatively high Mg status. Nevertheless, individuals with Mg deficiency might benefit from Mg^2+^ supplementation [67]. The main pathway of transcellular Mg^2+^ transport via enterocytes is constituted by the Mg^2+^-conducting chanzymes TRPM6/7, which mediate influx of Mg^2+^ into the cell and by the putative Na^+^/Mg^2+^ exchanger CNNM4 (cyclin and CBS domain divalent metal cation transport mediator 4) which is responsible for Mg^2+^ efflux [68,69,70,71]. According to profiles of protein expression, which are available in human protein atlas database (proteinatlas.org), also Na^+^/Mg^2+^ exchanger SLC41A1 (solute carrier family 41 member A1) is being abundantly expressed in gastro-intestinal tract. However, it is yet unknown, whether SLC41A1 plays as important role as CNNM4 in Mg^2+^ efflux in enterocytes, thus gut Mg^2+^ absorption [72].

#### Interaction between Magnesium and Intestinal Microbiota

Animal studies provide convincing evidence for an association between dietary Mg availability and microbial composition in the gut [73,74,75]. Jørgensen et al. [73] investigated the impact of dietary Mg deficiency on the composition of the gut microbiota in C57BL/6 mice. An Mg-deficient diet for 6 weeks altered the gut microbiota and was associated with altered anxiety-like behavior [73]. García-Legorreta et al. [76] studied the effect of low and high Mg diets on the modulation of the intestinal microbiota of male Wistar rats. Interestingly, after two weeks, the control and low Mg groups both had higher bacterial diversity than high Mg group. As the authors concluded for mice with no Mg deficiency, supplementation above recommended Mg intake levels can result in the development of intestinal dysbiosis. Hence, inadequate dietary Mg consumption may increase the capacity to harvest energy from the food [76].

Currently, little convincing evidence is available for an interaction between Mg deficiency and gut microbiota in humans. The combination of Mg^2+^ oxide with probiotics [Lactobacillus reuteri] appears to be promising for the treatment of chronic constipation in young patients. Interestingly, the administration of only *Lactobacillus reuteri* showed no significant decrease in stool consistency [77]. Magnesium also deserves attention in terms of bone density. Lambert et al. [78] studied postmenopausal osteopenic women supplemented with a combination of Mg, Ca, calcitriol, and bioavailable isoflavones and probiotics. Long-term consumption of minerals and probiotics for one year effectively affected bone metabolism and hormonal profile in estrogen deficient women. [78].

Intestinal bacteria may also play an important role in the bioavailability of Mg. Aljewitz et al. [22] observed higher bioaccessibility of Mg and other minerals from cheese in combination with probiotics. Cultures of *Lactobacillus spp.* ingested with Dutch-type cheese increased the availability of Mg (~18%) and Ca (~2.5%) in vitro [22]. Similarly, fermented goats’ milk containing *Lactobacillus plantarum* increased Mg bioavailability in comparison with commercial fermented goats’ milks [23]. Probiotics also increase the bioavailability of minerals in vegetable milks. Fermentation of soy milk reduced the content of nutrients such as phytic acid, thereby increasing the bioavailability of Mg and other minerals [79].

The recommended Mg intake might positively affect the composition of the intestinal microbiota and, consequently, the metabolism of the host, thus helping to prevent metabolic alterations associated with the development of metabolic syndrome and type 2 diabetes [80]. However, to improve our understanding of the effects of dietary Mg content on the human gut microbiome, we need additional clinical trials.

### 2.3. Iron

Fe (Fe) is the most abundant trace metal in the human body [81] and has a number of functions. It is particularly important to athletes, since it is indispensable for oxygen transport and the processes involved in energy metabolism [82]. Although Fe regulation and storage is essential for athletic performance, endurance athletes often lack it. Fe deficiency is three times more common among women than men [82]. However, Fe deficiency and anemia associated with Fe deficiency are not related to sport only but are considered global health issues leading to the deterioration in the quality of life of patients and often to a serious prognosis in patients with chronic diseases. The origin of Fe deficiency and anemia is usually a combination of an increased loss of Fe and a decrease in its intestinal absorption and delivery from Fe stores because of inflammation [83].

Dietary iron is present in form of ferritin, heme and inorganic Fe [84]. Iron from ferritin is being absorbed with the protein by yet unknown mechanism and liberated as Fe^2+^ in lysosomes (Figure 2A) [85]. Heme iron is internalized by endocytosis (Figure 2A). HCP1 (haem carrier protein 1) is being likely involved in this process. Heme iron is released as Fe^2+^ in endosomes (Figure 2A) [85]. The Fe-regulated ferrireductase DCYTB (duodenal cytochrome b) is responsible for reduction of non-heme Fe^3+^ to Fe^2+^, which is then transported via DMT1 (divalent metal transporter 1) into enterocytes [86] (Figure 2A). Iron in form of Fe^2+^ is being effluxed from enterocytes via FPN1 (ferroportin 1 or SLC40A1) (Figure 2A). Until present, FPN1 is the only known Fe^2+^ extruder [87].

Generally, supplementation in Fe-depleted patients optimizes and increases Fe absorption and might be a suitable therapeutic strategy [88]. Another approach is the fortification of cereals with ferrous and zinc sulfate as a strategy to prevent these deficiencies in the general population [89]. Nevertheless, “oral Fe is rife with unpleasant side-effects”, including constipation, gastric irritation, nausea, and a metallic taste [90]. Consumption of Fe supplements increases Fe in the large intestine and consequently affects the composition of gut microbiota by reducing the amount of lactic acid bacteria (*bifidobacteria* and *lactobacilli*) and increasing the entropathogenic *Escherichia coli*, which is associated with intestinal inflammation [91]. In vivo and in vitro studies have reported adverse effects of oral Fe supplementation on the composition of gut microbiota, on the gut metabolome, and on intestinal health in patients with chronic kidney disease, which in turn might result in an increased production of uremic toxins [92]. Shifts in the composition of gut bacteria have been associated with oral Fe replacement therapy in inflammatory bowel disease patients [93]. *Bifidobacteriaceae* seem promising in this context, because they are capable of binding Fe in the large intestine and of reducing the risk of colorectal cancer by limiting the formation of free radicals associated with Fe [24,94]. The activity of bacterial enzymes involved in colon carcinogenesis might be elevated by a high-meat Western-type diet. Supplements of *Lactobacillus acidophilus* can decrease these levels in both rats and humans [25]. Balamurugan et al. [95] have investigated if the intestinal microbiota of young anemic women in India differs from non-anemic women. In anemic women they found lower abundance of fecal Lactobacilli. However, no further differences were significant to any of the other examined bacteria. Of note, similar intakes of energy, carbohydrate, fiber, Fe, and milk were recorded in both groups [95].

Regarding previous, it is important to emphasize, that the molecular cross-talk between components of trans-cellular Fe absorption cascade and the microbiome is by far the best understood when compared to Ca, Mg, Zn and Se. On one hand, similar to Ca, Mg, and perhaps also other cations, SCFA facilitate absorption of Fe^2+^ in gut (Figure 2A) [84,96]. On the other hand very exciting work identifying two microbial metabolites in intestine, namely DAP (1,3-diaminopropane) and reuterin (3-hydroxypropionaldehyde, an antimicrobial compound produced by *Lactobacillus reuteri*), which inhibits ARNT (aryl hydro-carbon receptor nuclear translocator)–HIF-2α heterodimerization/translocation, thus downregulate expression of key components of iron transporst in enterocytes, *DMT1*, *DcytB* and *FPN* has been published recently by the group around Das et al. [97] (Figure 2B). Furthermore, the same group identified DAP and propionate as microbial metabolites capable of inhibiting HIF-2α expression (Figure 2B). Thus, Das and colleagues provided a direct evidence for involvement of microbial metabolite signaling in systemic Fe homeostasis.

#### Fermentation and Iron Bioavailability

Increasing the absorption of Fe from naturally occurring and unprocessed food may serve as an alternative to dietary supplements. The fermentation of cereals provides optimum pH conditions for the enzyme degradation of phytate, presented in the form of complexes with proteins and polyvalent cations such as zinc, calcium, magnesium, and Fe [98]. Phytate reduction might increase the levels of soluble and bioavailable minerals including Fe [99].

Another means of increasing the bioavailability of Fe is the soaking and germination of certain crops [100]. From this perspective, the plant microbiome, which includes the microbial community that typically interacts extensively with the plant, is of interest. The plant microbiome survives either inside or outside of plant tissues, performs numerous plant-beneficial activities, and promotes plant growth [101]. Teng et al. [102] have demonstrated that plant-derived exosome-like nanoparticles are taken up by the gut microbiota of animals and contain RNAs that alter microbiome composition and host physiology. Further research is urgently required if we are to understand the association between plant and human gut microbiomes and the mechanisms by which food products modulate commensals, a topic that remains basically unexplained. To better understand the effect of human intestinal microbiota on the bioavailability of Fe from plant foods, further research is needed. Cai et al. [103] suggest that human gut microbiota might increase the extraction of Fe from vegetables. They have recently shown, although only in vitro, that the bioaccessibility of Fe in the colon phase is probably significantly higher [1.3−1.8 times] than that in the small intestinal phase [103].

Dietary iron exists in two forms: heme iron (Fe^2+^) and non-heme iron (Fe^3+^) with transformation between these two states [104]. Fe^3+^ has to be reduced to the Fe^2+^ by a ferrireductase duodenal cytochrome before its uptake in the small intestine by the divalent metal transporter 1, which only transports Fe^2+^ [105]. In the anaerobic environment of the colon, Fe^3+^ can be easily reduced to Fe^2+^, leading to reductive dissolution of the Fe^3+^ compound [106]. Since Fe^3+^ is insoluble in most natural environments, many microbes rely on high-affinity Fe^3+^ siderophores for Fe^3+^ assimilation [107]. Luu and Ramsay [107] have proposed three mechanisms for the removal of iron from the siderophore: (a) the iron–siderophore complex is transported across the cell membrane, and the metal is released intracellularly because of low siderophore affinity for Fe^2+^; (b) the iron–siderophore complex is taken up into the cell, and iron is released through the destruction or chemical modification of the ligand; [c] the siderophore acts as an iron shuttle, but the iron does not enter the cell being instead donated to a second membrane-bound chelator followed by reduction, with the siderophore remaining outside cell [107].

When Fe supplementation is undertaken, the use of prebiotics and probiotics to mitigate the side effects of Fe on the gut should be taken in to account [93]. Probiotics seem to be effective with regard to Fe absorption in humans, as reported in the systematic review and meta-analysis of Vonderheid et al. [108]. They have found that the probiotic *Lactobacillus plantarum 299v* significantly increases non-heme dietary Fe absorption during a specific test period in crossover-designed studies compared with a control period [108].

### 2.4. Zinc

After iron, zinc [Zn] is the second most abundant trace element in the human body [81], with the majority of it occurring in skeletal muscles and bones [109]. However, dietary intake is required to prevent Zn deficiency, as the human body does not have a tissue depot for Zn [109]. Zn is an essential trace metal with crucial roles in growth, development, and the maintenance of immune function [110], plus catalytic, structural, and signaling functions affecting many cellular processes [109]. Zn deficiency has been associated with the immune response, cell proliferation, and pathogenesis and with the pathophysiology of selected diseases such as depression, bone disease, cardiovascular diseases, diabetes mellitus, Alzheimer’s disease, and Wilson’s disease [111,112]. Zn deficiency is relatively common in the physically active and in athletes [113]. In physically active individuals, Zn deficiency can lead to eating disorders characterized by weight loss, latent fatigue with decreased endurance capacity, and the risk of osteoporosis [114].

The nutritional habits that are often adopted in an attempt to enhance endurance performance, namely an excessive increase in carbohydrates and low intake of proteins and fat, can lead to inadequate Zn intake in 90% of athletes [114]. As concluded by Baranauskas et al. [115], particular attention should be focused on female athletes. These researchers evaluated the consumption of vitamins and minerals based on the dietary habits of 38 highly trained women. Barely one third consumed less Zn than the recommended daily intake [115]. A further group of higher risk individuals are those with low calorie intakes; they should consume foods with high micronutrient contents [116]. Special attention also needs to be paid to adult vegans and to physically active individuals [117]. In order to cover their daily recommended intake of Zn, athletes have to consume a well-balanced diet including supplements or fortified products [118]. However, no real benefit ensues from supplementation unless the athletes are Zn-deficient [113]. Consumption of Zn supplements has a limited, if any, effect on sports performance. [119]. The positive effects of Zn supplementation on the pathogenesis of COVID-19 are still questionable. Zn deficiency is possibly one of the factors predisposing individuals to infection and severe COVID-19, due to the fact that Zn is essential for the protection of natural tissue barriers such as respiratory epithelium [120]. Patients with severe COVID-19 are frequently also diagnosed as being Zn-deficient. This might result from modern dietary habits, which are often accompanied by Zn deficiency [121].

#### Association between Zinc and Gut Microbiota

Unlike iron, zinc is a divalent cation and does not require a redox reaction during the membrane transport process [122]. From more than twenty Zn transporters, Irt-related protein 4 is essential for the uptake of dietary zinc at the apical membrane in intestinal epithelial cells, and thus, the SLC39A4 gene mutation is associated with zinc deficiency and results in a rare inherited recessive disease (*Acrodermatitis enteropathica)* [123,124]. The Zn^2+^ efflux from enterocytes is secured by ZnT1 transporter (also known as SLC30A1), which may operate as Zn^2+^/H^+^ exchanger [125]. However, an insufficient dietary intake of Zn suggests the role of gut microbiota in this function, with the composition of the gut microbiota possibly affecting Zn absorption [126]. In vitro studies have indicated that the bioaccessibility of Zn from vegetables is mainly influenced by the microbiota of the colon [103,127]. However, the lower impact of the human gut microbiota on the bioaccessibility and bioavailability of elemental Zn from vegetables might be attributable to the bioaccessibility of Zn being higher in the small intestinal phase [103]. This is in accordance with the study of Intawongse and Dean [128] who have found that most of the metals coming from a wide range of vegetables are dissolved in the gastric and intestinal phases. Thus, gut microbiota might significantly affect the bioaccessibility of Zn contained in vegetables, as they reduce the dissolution of Zn in the colon phase. [103]. Based on the bioavailability of Zn, the best vegetable sources has been found to be lettuce. However, the consumption of 300 g of lettuce only meets 2.7% and 4.5% of Zn daily demands by males and females, respectively [103]. Currently, limited evidence coming mostly from animal studies has associated Zn with the gut microbiota [129,130,131]. Zn-deficient chicks have a significantly lower bacterial richness and α–diversity when measured on the Chao1 Index [130]. Zn deficiency in mice can negatively alter microbiota composition and function and gut-brain signaling and can trigger an increase of inflammatory markers [131].

Any Zn or Fe not absorbed in the small intestine of the healthy human reaches the colon and is available to colonocytes and/or commensal bacteria. Commensal bacteria probably increase the bioavailability of Zn and Fe and provide them to the host [132]. Under conditions of inflammation, pathogens outcompete commensal bacteria for these metals, thereby reducing Fe and Zn levels [132]. The consumption of Acacia, a prebiotic fiber, has been associated with a higher presence of *Lactobacillus* and *Bifidobacterium* spp in the gut and higher Zn concentrations in the femur of Wistar rats [26]. Four weeks after the consumption of gum arabic (a dried exudate of the acacia tree), the amount of *Bifidobacteria* and *Lactobacilli* was shown to increase in human stool samples, although Zn was not examined [133]. In another animal model, under heat stress conditions, zinc-enriched probiotics supplementation improved immune function by increasing serum IL-2, IL-6, and IFN-γ and decreasing IL-10 [28]. Contrary to benefits of Zn supplementation in deficient subjects, excessive dietary Zn in a mouse model altered the gut microbiota and decreased resistance to *C. difficile* infection [134].

**Table 1 ijms-22-06803-t001:** Interaction between gut microbiome and minerals.

Intervention	Study Population	Main Outcome	References
Soluble maize fiber	24 adolescent children (12–15 years)	Fractional Ca^2+^ absorption was 12% higher after treatment. Phylum Bacteroidetes was significantly greater	Whissner et al. [19]
Mg^2+^ oxide	60 young children with functional constipation (>6 month to <6 years)	Decrease in stool consistency and suppressed presence of the genus *Dialister*	Kubota et al. [77]
Iron sulfate	53 patients with IBD	Decreased abundances of *Ruminococcus bromii, Dorea sp.*, *Faecalibacterium prausnitzii and Collinsella aerofaciens*	Lee at a. [93]
Zn-biofortified wheat diet	Animal model (*Gallus gallus*)	Increased β-microbial diversity and increased Zn-dependent bacterial metabolic pathways	Reed at al. [129]
Se- and Zn-enriched *Lactobacillus plantarum*	Animal model (*Mus musculus*)	Increased antioxidant activity and blood Se level	Kang et al. [135]

### 2.5. Selenium

Selenium (Se) is an essential mineral in humans as it has antioxidant, anti-inflammatory, and immune functions, it participates in the metabolism of thyroid hormones, and it acts a cofactor of several selenoproteins [136,137]. Wardenaar et al. [138] examined whether high-level athletes followed the recommendations on micronutrients and whether their adequacy is linked to the use of nutritional and sports supplements. As expected, they found lower levels of Se in 10% of non-users [138]. One risk factor in deficiencies in Se might be veganism [139,140]. On the contrary, low-carbohydrate ketogenic diets preferably consumed by athletes for body composition benefits increase Se intake because of the nature of the consumed animal food [141]. However, supplementation with additional Se seems to provide limited beneficial effects on aerobic or anaerobic performance [142,143,144,145]. However, the association between Se levels and the risk of death from COVID-19 is noteworthy [146,147]. Se supplementation seems to have no effects on the athletic performance of physically active individuals with an adequate Se intake.

#### Interaction of Selenium and Probiotics

The gastrointestinal microbiota has been shown to affect Se status and selenoprotein expression in mice [148] and is able to sequester Se and to restrict its availability to the host [149]. Bacteria may compete with the host for Se when availability becomes limiting [148], although, in the mouse model, *Lactobacillus plantarum* enriched with Se and Zn increases antioxidant activity and blood Se levels [135]. In the study of Krausova et al. [27], the administration of Se-enriched lactic acid bacteria (*Streptococcus thermophilus* and *Enterococcus faecium*) improved the antioxidant status of the rats.

Food sources contain various chemical forms of bioselenocompounds, which might be transformed to selenomethionine [SeMet] by the gut microflora of a host, thereby allowing it to receive this essential micronutrient [150]. Takahashi et al. [150] suggest that SeMet is transported into bacterial cells; one part is incorporated into bacterial proteins, and the other part is excreted into the medium. SeMet-containing proteins are then available as a Se pool for the host [150]. Selenium-enriched *Bifidobacterium longum* from human stools efficiently biotransforms inorganic Se [Na_2_SeO_3_] into more bioactive organic Se forms (e.g., SeMet) [151]. In a study by Wastney et al. [152] the enteric absorption of single doses of SeMet by healthy subjects was 98%, indicating the high bioaccessibility of SeMet. Moreover, another investigation demonstrated that the bacterial selenized forms selenocystein (SeCys) and SeMet were also present in the liver and kidney tissues of rats after 58 days under an experimental diet [27]. In a further study, the Se content in the liver of animals fed a diet fortified with Se-enriched *Bifidobacterium longum* was much higher than that of animals fed a selenite-enriched diet [29]. Se- and Zn-enriched probiotics might thus be applied as promising functional food ingredients in the future [135]. Probiotics enriched with Se and Zn possibly increase bioavailability and absorption when compared with their inorganic forms, thus opening up another application for probiotics [153].

Beneficial effects of bacteria on absorbtion, bioaccessibility and bioavailability of all five discussed mineral are summarized in Table 2.

**Table 2 ijms-22-06803-t002:** The absorption, bioaccessibility, and bioavailability of essential minerals in relation to the gut microbiome.

Prebiotic/Probiotics	Mechanisms	Main Outcome	References
*Lactobacillus salivarius* and *Bifidobacterium infantis*	Transepithelial calcium transport	Enhanced intestinal calcium uptake	Gilman and Cashman [55]
Se-enriched *Bifidobacterium longum*	Biotransformation of inorganic Se into bioactive organic Se	High bioaccessibility of selenomethionine and 98% enteric absorbtion	Zhu et al. [151]; Wastney et al. [152]
Prebiotic fiber Acacia	Increased *Lactobacillus* and *Bifidobacterium spp* in the gut	Higher Zn concentrations in the femur of Wistar rats	Massot-Cladera et al. [26]
*Lactobacillus plantarum*	Microbial metabolite production, enhanced mucin production and immunomodulation	Increased non-heme dietary Fe absorption	Vonderheid et al. [26]
Soluble corn fiber, *Parabacteroides and Clostridium*	Acidification and SCFA production	Increased mineral solubility and calcium absorption	Trinidad et al. [26]; Cashman [44]
Fermented soymilk with various lactic acid bacteria	Reducing the content of phytic acid	Increasing the bioavailability of magnesium, calcium, iron and zinc	Rekha and Vijayalakshmi [79]
Fermented goats’ milks with *Lactobacillus plantarum*	Not totally clear	Increased magnesium and calcium bioavailability	Bergillos-Meca et al. [23]

## 3. Conclusions

The available literature indicates the beneficial effects of bacteria on mineral bioaccessibility and bioavailability and hence on their support of animal and human gut microbiomes and host vitality. The positive effect of probiotics on mineral absorption is promising in relation to the composition and metabolism of the gut microbiome. The fermentation of cereals and the soaking and germination of crops have the potential of providing suitable natural mineral supplements that can sustain both the human gut microbiome and overall health. Fermented foods and beverages might be helpful in increasing the levels of soluble and bioavailable micronutrients and might act as “nutrition supplements”. However, more research is needed if we are to improve our understanding of the role of microbiota in micronutrient metabolism and mechanisms related to the bioaccessibility and bioavailability of minerals in human gut.

## Figures and Tables

**Figure 1 ijms-22-06803-f001:**
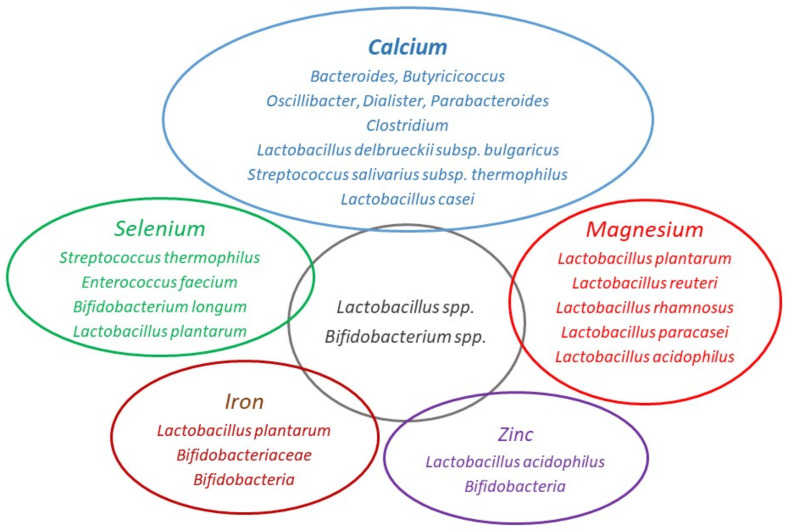
Bacterial strains related to mineral bioaccessibility and bioavailability. Image created according to Whisner et al. [19,20]; Amdekar et al. [21]; Aljewicz et al. [22]; Bergillos-Meca et al. [23]; Skrypnik and Suliburska [24]; Lidbeck et al. [25]; Massot-Cladera et al. [26]; Krausova et al. [27]; Malyar et al. [28] and Zhou et al. [29].

**Figure 2 ijms-22-06803-f002:**
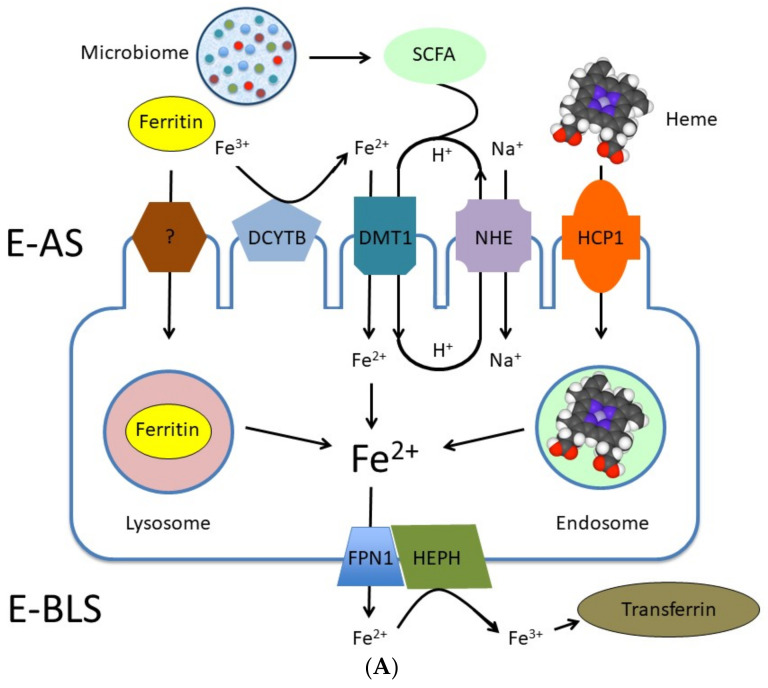
(**A**) Simplified model of iron transport via trans-cellular pathway in enterocytes. Dietary iron is absorbed on the apical side of enterocytes [E-AS] in form of ferritin, heme, or as Fe^2+^. A molecular background behind the uptake of ferritin by enterocytes is not yet clearly understood. Internalized ferritin undergoes degradation in lysosomes and Fe^2+^ is released. Endocytosis of heme is mediated by HCP1 (haem carrier protein 1). Heme is degraded in endosomes resulting in release of Fe^2+^. Nutritional iron has to be first reduced from Fe^3+^ to Fe^2+^ by ferrireductase DCYTB (duodenal cytochrome b) and it is uptaken as divalent cation by the enterocytes via DMT1 (divalent metal transporter 1), which operate in mode of Fe^2+^:H+ symporter. Recycling of H^+^ is maintained by NHE (Na^+^/H^+^ xchanger). SCFA (short chain fatty acids) produced by microbiota support the acidification on luminal side of the membrane. Fe^2+^ is exported from enterocytes on the basolateral side (E-BLS) by FPN1 (ferroportin 1). Released Fe2+ is oxidized by a transmembrane copper-dependent ferroxidase HEPH [hephaestin] to Fe3+, which is utilized by transferrin in circulation. Figure 2A was modified from Gulec et al. [85]. (**B**) Involvement of microbial metabolite signaling in systemic Fe homeostasis. Microbial metabolites DAP (1,3-diaminopropane) and reuterin (3-hydroxypropionaldehyde) inhibit ARNT (aryl hydro-carbon receptor nuclear translocator)–HIF-2α (transcription factor) heterodimerization/translocation. This inhibition results in downregulation of expression of DMT1, DcytB and FPN genes, which encode for the key components of Fe homeostasis and Fe^2+^ transport in enterocytes. Moreover, microbial metabolites DAP and propionate inhibit HIF-2α expression, thus influencing expression of Fe homeostatic factors/transporters.

## Data Availability

Not applicable.

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
