# Peer review of "Bioaccessibility and Bioavailability of Minerals in Relation to a Healthy Gut Microbiome"

_ijms, 2021, doi:10.3390/ijms22136803_

Round 1

Reviewer 1 Report

Viktor Bielik et al. proposed interesting reviews about associations between mineral absorption and gut microbiota.

In general, it comprehensively described the importance of mineral absorption for improving health, but it also raised concerns about mechanistic details regarding how gut microbiota can contribute the mineral absorption. Clues from the cited literature were mostly circumstantial based on small-scale observational studies, not directly suggesting possible mechanisms.

This reviewer agreed on the importance of bioaccessibility and bioavailability of minerals and contributions of gut microbiota. However, it did not provide sufficient mechanistic details to convince readers. For example, siderophore, two-component systems can be related to the mechanistic details (for example, see a paper like this (Emerging themes in manganese transport, biochemistry and pathogenesis in bacteria, FEMS Microbiology reviews, 2003)). This reviewer respectfully suggests improving the paper with more references to introduce possible mechanisms

Author Response

This reviewer agreed on the importance of bioaccessibility and bioavailability of minerals and contributions of gut microbiota. However, it did not provide sufficient mechanistic details to convince readers. For example, siderophore, two-component systems can be related to the mechanistic details (for example, see a paper like this (Emerging themes in manganese transport, biochemistry and pathogenesis in bacteria, FEMS Microbiology reviews, 2003)). This reviewer respectfully suggests improving the paper with more references to introduce possible mechanisms 

A: We agree with this point. We added more references (33 new references) describing mechanisms of absorption, bioaccessibility and bioavailability for each mineral and made a new summarizing table 2 and Fugures 1 and 2A, 2B. Changes are highlighted.

Reviewer 2 Report

This is a review article dealing with bioaccessibility and bioavailability of minerals in relation to a healthy gut microbiome. In general the manuscript is written with good English. However, there are several points the authors need to address before this paper could be accepted for publication. They are as follows:

  1. The authors have dealt with only iron, calcium, selenium and zinc. But what about other essential minerals?
  2. A new table incorporating the absorption, bioaccessibility and bioavailability data reported for essential minerals in relation to gut microbiome should be summarized.
  3. As shown for calcium, what is the effect of prebiotics and probiotics on other minerals?
  4. The authors should highlight on how the presence of one essential mineral affects the absorption, bioaccessibility and bioavailability of the other mineral in relation to gut microbiome.
  5. A new subtopic summarizing the effect of microencapsulated and nanoencapsulated minerals could affect the absorption, bioaccessibility and bioavailability in relation to gut microbiome.
  6. Some important figures (at least 3 to 5) from reported articles cited by the authors should be included.

Author Response

1.The authors have dealt with only iron, calcium, selenium and zinc. But what about other essential minerals?

A: We understand the reviewer's point. There are certainly more essential minerals that are of great importance for human composition, health and fitness. In this narrative review we discussed the bioaccessibility and bioavailability of major essential minerals that people commonly use as supplements (calcium, magnesium, iron, zinc and selenium). We agree, this point should be mentioned in the manuscript. We explained it in the Introduction.

2. A new table incorporating the absorption, bioaccessibility and bioavailability data reported for essential minerals in relation to gut microbiome should be summarized.

A: We made a new table (Table 2) according reviewer’s recommendations.

3. As shown for calcium, what is the effect of prebiotics and probiotics on other minerals?

A: We have improved the paper with more references about the effect of prebiotics and probiotics on each mineral.

4. The authors should highlight on how the presence of one essential mineral affects the absorption, bioaccessibility and bioavailability of the other mineral in relation to gut microbiome.

 A: We added additional text in revised manuscript.

5. A new subtopic summarizing the effect of microencapsulated and nanoencapsulated minerals could affect the absorption, bioaccessibility and bioavailability in relation to gut microbiome.

A: We agree about bioavailability of encapsulated minerals. Unfortunately to date there are limited data in scientific literature about gut microbiota and      encapsulated minerals. However we report some studies with encapsulated probiotics. This review is particular aimed to support bioavailability of naturally accessible minerals.

6. Some important figures (at least 3 to 5) from reported articles cited by the authors should be included.

A: Thank you for this suggestion. We added 3 more figures for better display.

Round 2

Reviewer 2 Report

The authors have satisfactorily addressed all the comments raised and therefore I recommend acceptance of this article for publication in IJMS.

Author Response

Thank you for all your comments. We believe that all revisions made will increase the quality and attractiveness for IJMS readers.